# Optimizing Bi-Encoder for Named Entity Recognition via Contrastive Learning

**Sheng Zhang, Hao Cheng, Jianfeng Gao, and Hoifung Poon**
Microsoft Research

## Abstract

We present a bi-encoder framework for named entity recognition (NER), which applies contrastive learning to map candidate text spans and entity types into the same vector representation space. Prior work predominantly approaches NER as sequence labeling or span classification. We instead frame NER as a representation learning problem that maximizes the similarity between the vector representations of an entity mention and its type. This makes it easy to handle nested and flat NER alike, and can better leverage noisy self-supervision signals. A major challenge to this bi-encoder formulation for NER lies in separating non-entity spans from entity mentions. Instead of explicitly labeling all non-entity spans as the same class `Outside` (O) as in most prior methods, we introduce a novel dynamic thresholding loss, learned in conjunction with the standard contrastive loss. Experiments show that our method performs well in both supervised and distantly supervised settings, for nested and flat NER alike, establishing new state of the art across standard datasets in the general domain (e.g., ACE2004, ACE2005, CoNLL2003) and high-value verticals such as biomedicine (e.g., GENIA, NCBI, BC5CDR, JNLPBA). We release the code at github.com/microsoft/binder.

## 1 Introduction

Named entity recognition (NER) is the task of identifying text spans associated with named entities and classifying them into a predefined set of entity types such as person, location, etc. As a fundamental component in information extraction systems (Nadeau & Sekine, 2007), NER has been shown to be of benefit to various downstream tasks such as relation extraction (Mintz et al., 2009), coreference resolution (Chang et al., 2013), and fine-grained opinion mining (Choi et al., 2006).

Inspired by recent success in open-domain question answering (Karpukhin et al., 2020) and entity linking (Wu et al., 2020; Zhang et al., 2021a), we propose an efficient **BI**-encoder for **NameD E**ntity **R**ecognition (**Binder**). Our model employs two encoders to separately map text and entity types into the same vector space, and it is able to reuse the vector representations of text for different entity types (or vice versa), resulting in a faster training and inference speed. Based on the bi-encoder representations, we propose a unified contrastive learning framework for NER, which enables us to overcome the limitations of popular NER formulations (shown in Figure 1), such as difficulty in handling nested NER with sequence labeling (Chiu & Nichols, 2016; Ma & Hovy, 2016), complex learning and inference for span-based classification (Yu et al., 2020; Fu et al., 2021), and challenges in learning with noisy supervision (Straková et al., 2019; Yan et al., 2021).[1] Through contrastive learning, we encourage the representation of entity types to be similar with the corresponding entity spans, and to be dissimilar with that of other text spans. Additionally, existing work labels all non-entity tokens or spans as the same class `Outside` (O), which can introduce false negatives when the training data is partially annotated (Das et al., 2022; Aly et al., 2021). We instead introduce a novel dynamic thresholding loss in contrastive learning, which learns candidate-specific dynamic thresholds to distinguish entity spans from non-entity ones.

To the best of our knowledge, we are the first to optimize bi-encoder for NER via contrastive learning. We conduct extensive experiments to evaluate our method in both supervised and distantly

---

[1]Das et al. (2022) applies contrastive learning for NER in a few-shot setting. In this paper, we focus on supervised NER and distantly supervised NER.

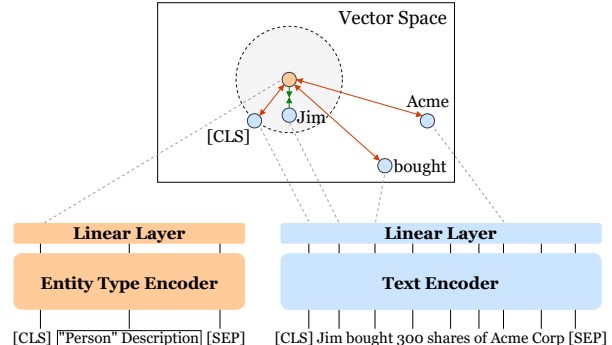

| | Nested NER | Noisy Supervision | Fast Speed |
|---|---|---|---|
| Sequence labeling | ✗ | ✗ | ✓ |
| Span-based classification | ✓ | ✗ | ✗ |
| Seq-to-seq generation | ✓ | ✗ | ✗ |
| **BINDER** (Ours) | ✓ | ✓ | ✓ |

Figure 1: Left: The architecture of **BINDER**. The entity type and text encoder are isomorphic and fully decoupled Transformer models. In the vector space, the anchor point (◯) represents the special token [CLS] from the entity type encoder. Through contrastive learning, we maximize the similarity between the anchor and the positive token (◯Jim), and minimize the similarity between the anchor and negative tokens. The dotted gray circle (delimited by the similarity between the anchor and ◯[CLS] from the text encoder) represents a threshold that separates entity tokens from non-entity tokens. To reduce clutter, data points that represent possible spans from the input text are not shown. Right: We compare **BINDER** with existing solutions for NER on three dimensions: 1) whether it can be applied to nested NER without special handling; 2) whether it can be trained using noisy supervision without special handling; 3) whether it has a fast training and inference speed.

supervised settings. Experiments demonstrate that our method achieves the state of the art on a wide range of NER datasets, covering both general and biomedical domains. In supervised NER, compared to the previous best results, our method obtains a 2.4%-2.9% absolute improvement in F1 on standard nested NER datasets such as ACE2004 and ACE2005, and a 1.2%-1.9% absolute improvement on standard flat NER datasets such as BC5-chem, BC5-disease, and NCBI. In distantly supervised NER, our method obtains a 1.5% absolute improvement in F1 on the BC5CDR dataset. We further study the impact of various choices of components in our method, and conduct breakdown analysis at entity type level and token level, which reveals potential growth opportunities.

## 2 METHOD

In this section, we present the design of **BINDER**, a novel architecture for NER tasks. As our model is built upon a bi-encoder framework, we first provide the necessary background for encoding both entity types and text using the Transformer-based (Vaswani et al., 2017) bi-encoder. Then, we discuss our ways of deriving entity type and individual mention span representations using the embedding output from the bi-encoder. Based on that, we introduce two types of contrastive learning objectives for NER using the token and span-level similarity respectively.

### 2.1 BI-ENCODER FOR NER

The overall architecture of **BINDER** is shown in Figure 1. Our model is built upon a bi-encoder architecture which has been mostly explored for dense retrieval (Karpukhin et al., 2020). Following the recent work, our bi-encoder also consists of two isomorphic and fully decoupled Transformer models (Vaswani et al., 2017), *i.e.* an entity type encoder and a text encoder. For NER tasks, we consider two types of inputs, entity type descriptions and text to detect named entities. At the high level, the entity type encoder produces type representations for each entity of interests (*e.g.* person in Figure 1) and the text encoder outputs representations for each input token in the given text where named entities are potentially mentioned (*e.g.* Jim in Figure 1). Then, we enumerate all span candidates based on corresponding token representations and match them with each entity type in the vector space. As shown in Figure 1, we maximize the similarity between the entity type and the positive spans, and minimize the similarity of negative spans.

We first formally discuss encoding both inputs using a pretrained Transformer model, BERT (Devlin et al., 2019).[2] Specifically, we use $x_1, \ldots, x_N$ to denote an input sequence of length $N$. When using BERT, there is a prepended token [CLS] and an appended token [SEP] for all input sequences, *i.e.* [CLS], $x_1, \ldots, x_N$ [SEP]. Then the output is a sequence of hidden states $\mathbf{h}_{[CLS]}, \mathbf{h}_1, \ldots, \mathbf{h}_N, \mathbf{h}_{[SEP]} \in \mathbb{R}^d$ from the last BERT layer for each input token, where $d$ is the hidden dimension. Note that as [CLS] is always in the beginning, $\mathbf{h}_0$ and $\mathbf{h}_{[CLS]}$ are interchangeable here. Based on this, we then discuss the way of computing entity type and text token embeddings, which are the basic building blocks for deriving our NER constrative learning objectives later.

**Entity Type Embeddings** The goal of entity type encoder is to produce entity type embeddings that serve as anchors in the vector space for contrastive learning. In this work, we focus on a predefined set of entity types $\mathcal{E} = \{E_1, \ldots, E_K\}$, where each entity type has one or multiple natural language descriptions. The natural language description can be *formal textual definitions* based on the dataset annotation guideline or Wikipedia, and *prototypical instances* where a target type of named entities are mentioned. For simplicity, the discussion proceeds with one description per type and we use $E_k$ to denote a sequence of tokens for the $k$-th entity type description. For a given entity type $E_k$, we use BERT as the entity type encoder ($\mathrm{BERT}^E$) and add an additional linear projection to compute corresponding entity type embeddings in the following way:

$$\mathbf{h}_{[CLS]}^{E_k} = \mathrm{BERT}^E(E_k), \tag{1}$$

$$\mathbf{e}_k = \mathrm{Linear}^E(\mathbf{h}_{[CLS]}^{E_k}), \tag{2}$$

where Linear is a learnable linear layer and $\mathbf{e}_k \in \mathbb{R}^d$ is the vector representation for $E_k$.

**Text Token Embeddings** Instead of using [CLS] embeddings as done in the recent bi-encoder work for entity retrieval (Wu et al., 2020), we consider using text token embeddings as the basic unit for computing similarity with entity span embeddings. As there are multiple potential named entities not known as a prior in the input, naively using special markers (Wu et al., 2020) incurs huge computation overhead for NER. Similar to the entity type embeddings, we again use BERT as the text encoder ($\mathrm{BERT}^T$) and simply use the final hidden states as the basic text token representations[3],

$$\mathbf{h}_1^T, \ldots, \mathbf{h}_N^T = \mathrm{BERT}^T(x_1, \ldots, x_N). \tag{3}$$

## 2.2 CONTRASTIVE LEARNING FOR NER

Based on the entity type embeddings and text token embeddings discussed above, we then introduce two different contrastive learning objectives for NER in this part. Here, we assume a *span* $(i, j)$ is a contiguous sequence of tokens in the input text with a start token in position $i$ and an end token in position $j$. Throughout this work, we use the similarity function, $\mathrm{sim}(\cdot, \cdot) = \frac{\cos(\cdot, \cdot)}{\tau}$, where $\tau$ is a scalar parameter.

As shown in Figure 1, the overall goal of NER constrative learning is to push the entity mention span representations close to their corresponding entity type embeddings (positive) and far away from irrelevant types (negative) in vector space, *e.g.* Person closer to Jim but away from Acme. To achieve that, we propose a multi-objective formulation consisting of two objectives based on *span* and *token* embedding spaces respectively.

**Span-based Objective** We derive the vector representation for span $(i, j)$ as below:

$$\mathbf{s}_{i,j} = \mathrm{Linear}^S(\mathbf{h}_i^T \oplus \mathbf{h}_j^T \oplus D(j - i)), \tag{4}$$

where $\mathbf{h}_i^T, \mathbf{h}_j^T$ are text token embeddings (Equation 3), $\mathbf{s}_{i,j} \in \mathbb{R}^d$, $\mathrm{Linear}^S$ is a learnable linear layer, $\oplus$ indicates the vector concatenation, $D(j - i) \in \mathbb{R}^m$ is the $(j - i)$-th row of a learnable span width embedding matrix $D \in \mathbb{R}^{N \times m}$. Based on this, the span-based infoNCE (Oord et al., 2018) can be defined as

$$\ell_{\mathrm{span}} = -\log \frac{\exp(\mathrm{sim}(\mathbf{s}_{i,j}, \mathbf{e}_k))}{\sum_{\mathbf{s}' \in \mathcal{S}_k^- \cup \mathbf{s}_{i,j}} \exp(\mathrm{sim}(\mathbf{s}', \mathbf{e}_k))}, \tag{5}$$

---

[2]Although different BERT variants are considered later in experiments, they all follow the same way of encoding discussed here.

[3]Here, we leave out special tokens for simplicity.

where the span $(i, j)$ belongs to entity type $E_k$, $\mathcal{S}_k^-$ is the set of negative spans that are all possible spans from the input text, excluding entity spans from $E_k$, and $\mathbf{e}_k$ is the entity type embedding.

**Position-based Objective**    One limitation of the span-based objective is that it penalizes all negative spans in the same way, even if they are partially correct spans, e.g., spans that have the same start or end token with the gold entity span. Intuitively, it is more desirable to predict partially correct spans than completely wrong spans. Therefore, we propose additional position-based contrastive learning objectives. Specifically, we compute two additional entity type embeddings for $E_k$ by using additional linear layers, $e_k^B = \texttt{Linear}_B^E(\mathbf{h}_{[\texttt{CLS}]}^{E_k})$, $e_k^Q = \texttt{Linear}_Q^E(\mathbf{h}_{[\texttt{CLS}]}^{E_k})$, where $e_k^B, e_k^Q$ are the type embeddings for the start and end positions respectively, $\mathbf{h}_{[\texttt{CLS}]}^{E_k}$ is from the entity type encoder (Equation 1). Accordingly, we can use two addtional linear layers to compute the corresponding token embeddings for the start and end tokens respectively, $\mathbf{u}_n = \texttt{Linear}_B^T(\mathbf{h}_n^T)$, $\mathbf{v}_n = \texttt{Linear}_Q^T(\mathbf{h}_n^T)$, where $\mathbf{h}_n^T$ is the text token embeddings (Equation 3). Using $e_B^k, e_Q^k$ as anchors, we then define two position-based objectives via contrastive learning:

$$\ell_{\text{start}} = -\log \frac{\exp(\text{sim}(\mathbf{u}_i, \mathbf{e}_k^B))}{\sum_{\mathbf{u}' \in \mathcal{U}_k^- \cup \mathbf{u}_i} \exp(\text{sim}(\mathbf{u}', \mathbf{e}_k^B))} \tag{6}$$

$$\ell_{\text{end}} = -\log \frac{\exp(\text{sim}(\mathbf{v}_j, \mathbf{e}_k^Q))}{\sum_{\mathbf{v}' \in \mathcal{V}_k^- \cup \mathbf{v}_j} \exp(\text{sim}(\mathbf{v}', \mathbf{e}_k^Q))}, \tag{7}$$

where $\mathcal{U}_k^-, \mathcal{V}_k^-$ are two sets of positions in the input text that do not belong to the start/end of any span of entity type $k$. Compared with Equation 5, the main difference of position-based objectives comes from the corresponding negative sets where start and end positions are independent of each other. In other words, the position-based objectives can potentially help the model to make better start and end position predictions.

**Thresholding for Non-Entity Cases**    Although the contrastive learning objectives defined above can effectively push the positive spans close to their corresponding entity type in vector space, it might be problematic for our model at test time to decide how close a span should be before it can be predicted as positive. In other words, the model is not able to separate entity spans from non-entity spans properly. To address this issue, we use the similarity between the special token $[\texttt{CLS}]$ and the entity type as a dynamic threshold (as shown in Figure 1). Intuitively, the representation of $[\texttt{CLS}]$ reads the entire input text and summarizes the contextual information, which could make it a good choice to estimate the threshold to separate entity spans from non-entity spans. We compare it with several other thresholding choices in §4.

To learn thresholding, we extend the original contrastive learning objectives with extra adaptive learning objectives for non-entity cases. Specifically, for the start loss (Equation 6), the augmented start loss $\ell_{\text{start}}^+$ is defined as

$$\ell_{\text{start}}^+ = \beta \ell_{\text{start}} - (1 - \beta) \log \frac{\exp(\text{sim}(\mathbf{u}_{[\texttt{CLS}]}, \mathbf{e}_k^B))}{\sum_{\mathbf{u}' \in \mathcal{U}_k^-} \exp(\text{sim}(\mathbf{u}', \mathbf{e}_k^B))}. \tag{8}$$

An augmented end loss $\ell_{\text{end}}^+$ can be defined in a similar fashion. For the span loss (Equation 5), we use the span embedding $\mathbf{s}_{0,0}$ for deriving the augmented span loss $\ell_{\text{span}}^+$

$$\ell_{\text{span}}^+ = \beta \ell_{\text{span}} - (1 - \beta) \log \frac{\exp(\text{sim}(\mathbf{s}_{0,0}, \mathbf{e}_k))}{\sum_{\mathbf{s}' \in \mathcal{S}_k^-} \exp(\text{sim}(\mathbf{s}', \mathbf{e}_k))}. \tag{9}$$

Note that we use a single scalar hyperparameter $\beta$ for balancing the adaptive thresholding learning and original contrastive learning for all three cases.

**Training**    Finally, we consider a multi-task contrastive formulation by combing three augmented contrastive learning discussed above, leading to our overall training objective

$$\mathcal{L} = \alpha \ell_{\text{start}}^+ + \gamma \ell_{\text{end}}^+ + \lambda \ell_{\text{span}}^+, \tag{10}$$

where $\alpha, \gamma, \lambda$ are all scalar hyperparameters.

**Inference Strategy** During inference, we enumerate all possible spans that are less than the length of $L$ and compute three similarity scores based on the start/end/span cases for each entity type. We consider two prediction strategies, *joint position-span* and *span-only* predictions. In the joint position-span case, for entity type $E_k$, we prune out spans $(i, j)$ that have either start or end similar scores lower than the learned threshold, *i.e.* $\text{sim}(\mathbf{u}_i, \mathbf{e}_k^{\text{B}}) < \text{sim}(\mathbf{u}_{\texttt{[CLS]}}, \mathbf{e}_k^{\text{B}})$ or $\text{sim}(\mathbf{v}_j, \mathbf{e}_k^{\text{Q}}) < \text{sim}(\mathbf{v}_{\texttt{[CLS]}}, \mathbf{e}_k^{\text{Q}})$. Then, only those spans with span similarity scores higher than the span threshold are predicted as positive ones *i.e.* $\text{sim}(\mathbf{s}_{i,j}, \mathbf{e}_k) > \text{sim}(\mathbf{s}_{0,0}, \mathbf{e}_k)$. For the span-only strategy, we just rely on the span similarity score and keep all qualified spans as final predictions. As shown later in our experiments (§4), we find the span-only inference is more effective, because the joint inference is more likely to be affected by annotation artifacts. The full inference algorithm is summarized in Appendix A.5.

## 3 EXPERIMENTS

We evaluate our method in both supervised and distantly supervised settings. The implementation details of our method are described in Appendix A.4

**Evaluation Metrics** We follow the standard evaluation protocol and use micro F1: a predicted entity span is considered correct if its span boundaries and the predicted entity type are both correct.

**Datasets** In the *supervised setting*, we evaluate our method on both nested and flat NER. For nested NER, we consider ACE2004, ACE2005, and GENIA (Kim et al., 2003). ACE2004 and ACE2005 are collected from general domains (e.g., news and web). We follow Luan et al. (2018) to split ACE2004 into 5 folds, and ACE2005 into train, development and test sets. GENIA is from the biomedical domain. We use its v3.0.2 corpus and follow Finkel & Manning (2009) and Lu & Roth (2015) to split it into 80%/10%/10% train/dev/test splits. For flat NER, we consider CoNLL2003 (Tjong Kim Sang & De Meulder, 2003) as well as five biomedical NER datasets from the BLURB benchmark (Gu et al., 2021): BC5-chem/disease (Li et al., 2016), NCBI (Doğan et al., 2014), BC2GM (Smith et al., 2008), and JNLPBA (Collier & Kim, 2004). Preprocessing and splits follow Gu et al. (2021). Appendix A.6 reports the dataset statistics.

In the *distantly supervised setting*, we consider BC5CDR (Li et al., 2016). It consists of 1,500 articles annotated with 15,935 chemical and 12,852 disease mentions. We use the standard train, development, and test splits. In the train and development sets, we discard all human annotations and only keep the unlabeled articles. Their distant labels are generated using exact string matching against a dictionary released by Shang et al. (2018).[4] On the training set, the distant labels have high precision (97.99% for chemicals, and 98.34% for diseases), but low recall (63.14% for chemicals, and 46.73% for diseases).

**Supervised NER Results** Table 1 presents the comparison of our method with all previous methods evaluated on three nested NER datasets, ACE2004, ACE2005, and GENIA. We report precision, recall, and F1. As is shown, our method achieves the state of the art performance on all three datasets. On ACE2004 and ACE2005, it outperforms all previous methods with 89.7% and 90.0% F1. Particularly, in comparison with the previous best method (Tan et al., 2021), our method significantly improves F1 by the absolute points of +2.4% and +2.9% respectively. On GENIA, our method advances the previous state of the art (Yu et al., 2020) by +0.3% F1. Note that the previous methods are built on top of different encoders, e.g., LSTM, BERT-large, BART-large, and T5-base. We also report our method using a BERT-base encoder, which even outperforms the previous methods that use a larger encoder of BERT-large (Tan et al., 2021) or BART-large (Yan et al., 2021). Overall, our method has substantial gains over the previous methods. Due to the space limit, we report the result on CoNLL03 and compare with prompt-based NER and span-based NER in Appendix A.1 and A.2.

Table 2 compares our method with all previous submissions on the BLURB benchmark. We only report F1 due to unavailability of precision and recall of these submissions. The major difference among these submissions are encoders. A direct comparison can be made between our method and Gu et al. (2021) which formulates NER as a sequential labeling task and fine-tunes a standard LSTM+CRF classifier on top of the pretrained transformer encoder. While both using PubMedBERT

---

[4]github.com/shangjingbo1226/AutoNER

as the encoder, our method significantly outperforms Gu et al. (2021) across the board. Compared to the previous best submissions (Kanakarajan et al., 2021; Yasunaga et al., 2022), our method also show substantial gains on BC5-chem (+1.2%), BC5-disease (+1.9%), and NCBI (1.5%).

Table 1: Test scores on three nested NER datasets. **Bold-and-underline**, **bold-only** and underline-only indicate the best, the second best, and the third best respectively. The different encoders are used: L = LSTM, Bl = BERT-large, BioB = BioBERT, BioBl = BioBERT-large, BAl = BART-large, T5b = T5-base, Bb = BERT-base. † Original scores of Li et al. (2020) are not reproducible. Like Yan et al. (2021), we report the rerun of their released code.

| | Encoder | ACE2004 | | | ACE2005 | | | GENIA | | |
|---|---|---|---|---|---|---|---|---|---|---|
| | | P | R | F1 | P | R | F1 | P | R | F1 |
| Lu & Roth (2015) | - | 70.0 | 56.9 | 62.8 | 66.3 | 59.2 | 62.5 | 74.2 | 66.7 | 70.3 |
| Katiyar & Cardie (2018) | L | 73.6 | 71.8 | 72.7 | 70.6 | 70.4 | 70.5 | 77.7 | 71.8 | 74.6 |
| Shibuya & Hovy (2020) | Bl | 83.7 | 81.9 | 82.8 | 83.0 | 82.4 | 82.7 | 78.1 | 76.5 | 77.3 |
| Wang et al. (2020) | Bl/BioB | 86.1 | 86.5 | 86.3 | 84.0 | 85.4 | 84.7 | 79.5 | 78.9 | 79.2 |
| Li et al. (2020)† | Bl/BioB | 85.8 | 85.8 | 85.8 | 85.0 | 84.1 | 84.6 | 81.2 | 76.4 | 78.7 |
| Yu et al. (2020) | Bl/BioB | 87.3 | 86.0 | 86.7 | 85.2 | 85.6 | 85.4 | 81.8 | 79.3 | **80.5** |
| Tan et al. (2021) | Bl/BioB | 88.5 | 86.1 | 87.3 | 87.5 | 86.6 | 87.1 | 82.3 | 78.7 | 80.4 |
| Yan et al. (2021) | BAl | 87.3 | 86.4 | 86.8 | 83.2 | 86.4 | 84.7 | 78.6 | 79.3 | 78.9 |
| Zhang et al. (2022) | T5b | 86.5 | 84.5 | 85.4 | 83.3 | 86.6 | 84.9 | 81.0 | 77.2 | 79.1 |
| Wan et al. (2022) | Bb | 86.7 | 85.9 | 86.3 | 84.4 | 85.9 | 85.1 | 77.9 | 80.7 | 79.3 |
| **BINDER** (Ours) | Bb/BioB | 88.3 | 89.1 | **88.7** | 89.1 | 89.8 | **89.5** | 81.5 | 79.6 | **80.5** |
| | Bl/BioBl | 89.7 | 89.7 | **89.7** | 89.6 | 90.5 | **90.0** | 83.4 | 78.3 | **80.8** |

Table 2: Test F1 scores on five flat NER datasets from the BLURB benchmark (aka.ms/blurb). Bold and underline indicate the best and the second best respectively. All encoders use their base version.

| | Encoder | BC5-chem | BC5-disease | NCBI | BC2GM | JNLPBA |
|---|---|---|---|---|---|---|
| Lee et al. (2019) | BioBERT | 92.9 | 84.7 | 89.1 | 83.8 | 78.6 |
| Gu et al. (2021) | PubMedBERT | 93.3 | 85.6 | 87.8 | 84.5 | 79.1 |
| Kanakarajan et al. (2021) | BioELECTRA | 93.6 | 85.8 | 89.4 | 84.7 | 80.2 |
| Yasunaga et al. (2022) | LinkBERT | 93.8 | 86.1 | 88.2 | **84.9** | 79.0 |
| **BINDER** (Ours) | PubMedBERT | **95.0** | **88.0** | **90.9** | 84.6 | **80.3** |

**Distantly Supervised NER Results** Table 3 compares test scores of our method and previous methods on BC5CDR. It presents a clear advantage of our method over all previous methods in the distantly supervised setting. The F1 score is advanced by +1.5% over the previous best method (Zhou et al., 2022), which adapts positive and unlabeled (PU) learning to obtain a high recall yet at the loss of precision. In contrast, our method maintains a reasonable recall (comparable to Shang et al., 2018; Peng et al., 2019) and substantially improves the precision. Note that besides the dictionary used to generate distant supervisions, Zhou et al. (2022); Shang et al. (2018) require an additional high-quality dictionary to estimate the noisiness of non-entity spans. Our method does not have such a requirement. For reference, Table 2 also includes the supervised state of the art. Our method in the supervised setting achieves 91.9% F1, outperforming the previous SOTA Wang et al. (2021) by 1.0%. Comparing both settings, we observe that the distantly supervised result still has an over-10-point gap with the supervised one, indicating a potential to further reduce the false negative noise.

Table 3: Test scores on BC5CDR. All baselines scores in the distantly supervised setting are quoted from Zhou et al. (2022).

| | BC5CDR | | |
|---|---|---|---|
| | P | R | F1 |
| *Distantly Supervised* | | | |
| Dict/KB Matching | 86.4 | 51.2 | 64.3 |
| AutoNER (Shang et al., 2018) | 82.6 | 77.5 | 80.0 |
| BNPU (Peng et al., 2019) | 48.1 | 77.1 | 59.2 |
| BERT-ES (Liang et al., 2020) | 80.4 | 67.9 | 73.7 |
| Conf-MPU (Zhou et al., 2022) | 76.6 | 83.8 | 80.1 |
| **BINDER** (Ours) | 87.6 | 76.3 | **81.6** |
| *Fully Supervised* | | | |
| Nooralahzadeh et al. (2019) | 92.1 | 87.9 | 89.9 |
| Wang et al. (2021) | - | - | 90.9 |
| **BINDER** (Ours) | 92.6 | 91.2 | **91.9** |

## 4 ANALYSIS

Here we conduct extensive analyses of our method. For efficiency, all analysis in this section is done based on the uncased BERT-base encoder.

**Ablation Study**  We compare several variants of our method and report their test scores on ACE2005 in Table 4. We observe performance degradation in all these variants. *Shared linear layers* uses the same linear layer for span and token embeddings, and the same linear layer for entity type embeddings, in the hope of projecting them into the same vector space and sharing their semantics. It leads to a slightly better precision but a drop of recall. Similar changes are observed in *Joint position-span inference*, which adopts a more stringent strategy to prune out spans – only keep spans whose start, end, and span scores are all above the thresholds. *No position-based objectives* only optimizes the span-based objective, which penalizes partially corrected spans in the same way as other spans, resulting in a marginal improvement of recall but a significant loss of precision.

**Choices of Entity Type Descriptions**  Table 5 compares different choices of entity type descriptions. Our final model uses annotation guidelines, which outperforms other choices: (1) *Atomic labels* considers each entity type as an atomic label. Instead of learning an entity type encoder, we directly learn an embedding vector for each entity type. (2) *Keywords* uses a keyword for each entity type as the input to the entity type encoder, e.g., "person" for PER. (3) *Prototypical instances* for each minibatch during training dynamically samples from the training data a prototypical instance for each entity type and uses it as input to the entity type encoder. Unlike annotation guidelines, we add special markers to indicate the start and end of an entity span and use the hidden states of the start marker as entity type embeddings. At test time, we increase the number of prototypical instances to three for each entity type. Larger number of prototypical instances may improve the performance. Prototypical instances may also lead to a better performance in few-shot or zero-shot settings. We leave them to future exploration.

Table 4: Test scores of our method and its variants on ACE2005.

Table 5: Test scores on ACE2005 with different entity type descriptions.

| | **ACE2005** | | |
|---|---|---|---|
| | P | R | F1 |
| Our full model | 89.1 | 89.8 | 89.5 |
| Shared linear layers | 89.3 | 89.3 | 89.3 |
| Joint position-span inference | 89.4 | 89.2 | 89.3 |
| No position-based objectives | 88.7 | 89.9 | 89.3 |

| | ACE2005 | | |
|---|---|---|---|
| | P | R | F1 |
| Atomic labels | 88.9 | 89.6 | 89.2 |
| Keywords | 88.7 | 89.8 | 89.2 |
| Prototypical instances | 88.7 | 90.1 | 89.4 |
| Annotation guidelines | 89.1 | 89.8 | 89.5 |

**Thresholding Strategies**  Our method uses dynamic similarity thresholds to distinguish entity spans from non-entity spans. We compare the impact of dynamic thresholds in our method with two straightforward variants: (1) *Learned global thresholds* replaces dynamic thresholds with global thresholds, one for each entity type. Specifically, we consider the global similarity thresholds as scalar parameters (initialized as 0). During training, we replace the similarity function outputs $\text{sim}(\mathbf{u}_{\texttt{[CLS]}}, \mathbf{e}_k^{\text{B}})$ in Equation 8 and $\text{sim}(\mathbf{s}_{0,0}, \mathbf{e}_k)$ in Equation 9 with the corresponding global thresholds. At test times, the global thresholds are used to separate entity spans from non-entity spans. (2) *Global thresholds tuned on dev* introduces global thresholds after the training is done and tune them on the development set. During training, instead of Equation 10, we optimize a simplified loss without thresholding, $\alpha\ell_{\texttt{start}} + \gamma\ell_{\texttt{end}} + \lambda\ell_{\texttt{joint}}$. Table 6 compares their test scores on the ACE2005 dataset. Dynamic thresholds have the best scores overall. Learned global thresholds performs better than global thresholds tuned on the development set, indicating the necessity of learning thresholds during training. Note that the global thresholds tuned on dev still outperforms all the previous methods in Table 1, showing a clear advantage of our bi-encoder framework. In Appendix A.7, We further visualize and discuss the distribution of similarity scores between text spans and entity types based on different thresholding strategies.

**Time Efficiency**  Table 7 compares the training and inference speed of our method against several prominent methods. To ensure fair comparisons, all speed numbers are recorded based on the same machine using their released code with the same batch size and the same maximum sequence length.

MRC-NER (Li et al., 2020) formulates NER as a machine reading comprehension problem and employs a cross-attention encoder. Comparing with it, our bi-encoder framework has 17x and 8x speed on training and inference respectively. Biaffine-NER (Yu et al., 2020) formulates NER as a dependency parsing problem and applies a biaffine classifier to classify the typed arc between an entity span start and end. Comparing with it, our framework does not need a biaffine layer and is 1.7x and 2.4x faster at training and inference. Without the need of conditional random fields (CRFs), our framework is even faster than the widely used BERT-CRF framework. A drawback of BERT-CRF is that it cannot be applied to nested NER. Here we test it on a flat NER dataset CoNLL2003 (Tjong Kim Sang & De Meulder, 2003). All other frameworks are tested on ACE2005.

Table 6: Test scores of our method using different thresholding strategies on ACE2005.

|  | ACE2005 | | |
|---|---|---|---|
|  | P | R | F1 |
| Dynamic thresholds | 89.1 | 89.8 | 89.5 |
| Learned global thresholds | 88.2 | 89.0 | 88.6 |
| Global thresholds tuned on dev | 86.3 | 88.7 | 87.5 |

Table 7: Training and inference speed.

|  | Speed (w/s) | |
|---|---|---|
|  | Training | Inference |
| MRC-NER (Li et al., 2020) | 147 | 1,110 |
| Biaffine-NER (Yu et al., 2020) | 1,548 | 3,634 |
| BERT-CRF (pytorch_neural_crf) | 2,273 | 8,596 |
| **BINDER** (Ours) | **2,571** | **8,886** |

Table 8: Test F1 score breakdowns on ACE2005 and GENIA. Columns compare F1 scores on different entity types. Rows compare F1 scores based on the entire entity span, or only the start or end of entity span. S-F1 denotes the strict F1 requiring the exact boundary match. L-F1 denotes the loose F1 allowing partial overlaps. The color signifies substantially better F1 scores than the corresponding entity span strict F1 scores.

|  | ACE2005 | | | | | | | | GENIA | | | | | |
|---|---|---|---|---|---|---|---|---|---|---|---|---|---|---|
|  | PER | GPE | ORG | FAC | LOC | VEH | WEA | ALL | Prot. | DNA | CellT. | CellL. | RNA | ALL |
| $\text{S-F1}_{\text{span}}$ | 93.4 | 91.2 | 79.7 | 81.0 | 78.7 | 84.8 | 82.1 | 89.5 | 82.9 | 77.6 | 74.5 | 76.3 | 87.9 | 80.5 |
| $\text{S-F1}_{\text{start}}$ | 93.9 | 91.2 | 80.7 | 81.0 | 79.0 | 84.8 | 82.1 | 89.9 | 86.1 | 80.9 | 74.5 | 80.2 | 88.7 | 83.2 |
| $\text{S-F1}_{\text{end}}$ | 93.9 | 91.2 | 81.9 | 83.1 | 79.0 | 86.8 | 82.1 | 90.3 | 87.6 | 82.6 | 83.7 | 82.8 | 91.0 | 85.8 |
| $\text{L-F1}_{\text{span}}$ | 94.4 | 91.4 | 83.0 | 83.1 | 79.4 | 87.2 | 82.1 | 90.8 | 91.6 | 87.4 | 84.8 | 87.2 | 91.7 | 89.9 |

**Performance Breakdown** Table 8 shows the test F1 scores on each entity type of ACE2005 and GENIA. We report four types of F1 scores: $\text{S-F1}_{\text{span}}$ is the strict F1 based on the exact match of *entity spans*; $\text{S-F1}_{\text{start}}$ is the strict F1 based on the exact match of *entity start words*; $\text{S-F1}_{\text{end}}$ is the strict F1 based on the exact match of *entity end words*; $\text{L-F1}_{\text{span}}$ is the loose F1 allowing the *partial* match of entity spans. We notice that $\text{S-F1}_{\text{end}}$ is often substantially better than $\text{S-F1}_{\text{span}}$ and $\text{S-F1}_{\text{start}}$. To explain this difference, we go through these partially corrected predictions and summarize the common errors in Table 9. The most common one is the inclusion or exclusion of modifiers. This could be due to annotation disagreement: in ACE2005, sometimes generic spans are preferred (e.g., "tomcats" vs. "f-14 tomcats"), while in some cases specific spans are preferred (e.g., "cruise ship" vs. "ship"). This issue is more common in the biomedical dataset GENIA, e.g., "human GR" is considered as wrong while "human lymphocytes" are correct, which explains the higher scores of $\text{S-F1}_{\text{end}}$ than $\text{S-F1}_{\text{start}}$. Missing genitives for person names are another common errors in ACE2005. We also discover some annotation errors, where names of a single person, protein, or cell line are sometimes broken into two less meaningful spans, e.g., "Dr. Germ" is annotated as two person mentions "Dr"

Table 9: Examples of common errors among the partially corrected predictions. Red indicates error spans. Blue indicates missing spans. The number after each span mean the span frequency in the training data.

| Error Type | Ent. Type | Predicted ↔ Gold |
|---|---|---|
| **Modifier Error** | VEH | f-14 tomcats (0) ↔ tomcats (0) |
|  | FAC | federal court (0) ↔ court (0) |
|  | VEH | ship (29) ↔ cruise ship (1) |
|  | CellL. | unstimulated T cells (0) ↔ T cells (553) |
|  | Prot. | human GR (0) ↔ GR (88) |
|  | CellT. | lymphocytes (117) ↔ human lymphocytes (18) |
|  | DNA | E6 motif (0) ↔ synthetic E6 motif (0) |
| **Missing Genitive** | PER | attendant (3) ↔ attendant's (0) |
| **Annotation Error** | PER | Dr. Germ (0) ↔ Dr (1) / Germ (0) |
|  | Prot. | Ag amino acid sequence (0) ↔ Ag (1) / amino acid sequence (6) |
|  | CellL. | EBV-transformed human B cell line SKW6.4 (0) ↔ EBV-transformed human B cell line (0) / SKW6.4 (1) |
|  | DNA | second-site LTR revertants (0) ↔ second-site LTR (0) |

and "Germ", and "EBV-transformed human B cell line SKW6.4" is annotated as two separate cell lines "EBV-transformed human B cell line" and "SKW6.4".

## 5 RELATED WORK

**Supervised NER** Early techniques for NER are based on hidden markov models (e.g., Zhou & Su, 2002) and conditional random fields (CRFs) (e.g., McDonald & Pereira 2005). However, due to the inability to handle nested named entities, techniques such as cascaded CRFs (Alex et al., 2007), adpated constituency parsing (Finkel & Manning, 2009), and hypergraphs (Lu & Roth, 2015) are developed for nested NER. More recently, with the advance in deep learning, a myriad of new techniques have been used in supervised NER. Depending on the way they formulate the task, these techniques can be categorized as NER as sequence labeling (Chiu & Nichols, 2016; Ma & Hovy, 2016; Katiyar & Cardie, 2018); NER as parsing (Lample et al., 2016; Yu et al., 2020); NER as span classification (Sohrab & Miwa, 2018; Xia et al., 2019; Ouchi et al., 2020; Fu et al., 2021); NER as a sequence-to-sequence problem (Straková et al., 2019; Yan et al., 2021); and NER as machine reading comprehension (Li et al., 2020; Mengge et al., 2020). Unlike previous work, we formulate NER as a contrastive learning problem. The span-based design of our bi-encoder and contrastive loss provides us with the flexibility to handle both nested and flat NER.

**Distantly Supervised NER** Distant supervision from external knowledge bases in conjunction with unlabeled text is generated by string matching (Giannakopoulos et al., 2017) or heuristic rules (Ren et al., 2015; Fries et al., 2017). Due to the limited coverage of external knowledge bases, distant supervision often has a high false negative rate. To alleviate this issue, Shang et al. (2018) design a new tagging scheme with an `unknown` tag specifically for false negatives; Mayhew et al. (2019) iteratively detect false negatives and downweigh them in training; Peng et al. (2019); Zhou et al. (2022) address overfitting to false negatives using Positive and Unlabeled (PU) learning; Zhang et al. (2021b) identify dictionary biases via a structural causal model, and de-bias them using causal interventions. Liu et al. (2021) introduce a calibrated confidence estimation method and integrate it into a self-training framework. Without replying on sophisticated de-noising designs, our bi-encoder framework can be directly used in distant supervision. Experiments in §3 show the robustness of our contrastive learning algorithm to the noise in distantly supervised NER.

**Bi-Encoder** The use of bi-encoder dates back to Bromley et al. (1993) for signature verification and Chopra et al. (2005) for face verification. These works and their descendants (e.g., Yih et al., 2011; Hu et al., 2014) refer to the architecture as *siamese networks* since two similar inputs are encoded separately by two copies of the same network (all parameters are shared). Wsabie (Weston et al., 2010) and StarSpace (Wu et al., 2018) subsequently employ bi-encoder to learn embeddings for different data types. Using deep pretrained transformers as encoders, Humeau et al. (2019) compare three different architectures, bi-encoder, poly-encoder, and cross-encoder. The bi-encoder architecture afterwards has been use in various tasks, e.g., information retrieval (Huang et al., 2013; Gillick et al., 2018), open-domain question answering (Karpukhin et al., 2020), and entity linking (Gillick et al., 2019; Wu et al., 2020; Zhang et al., 2021a). To our best knowledge, there is no previous work learning bi-encoder for NER.

## 6 CONCLUSIONS

We present a bi-encoder framework for NER using contrastive learning, which separately maps text and entity types into the same vector space. To separate entity spans from non-entity ones, we introduce a novel contrastive loss to jointly learn span identification and entity classification. Experiments in both supervised and distantly supervised settings show the effectiveness and robustness of our method. We conduct extensive analysis to explain the success of our method and reveal growth opportunities. Future directions include further improving low-performing types and applications in self-supervised zero-shot settings.

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

## A APPENDIX

### A.1 COMPARISON WITH PROMPT-BASED LEARNING

Prompt-based learning emerges as a powerful method in few-shot NER (Cui et al., 2021; Ma et al., 2022; Ding et al., 2021). Below, we compare **BINDER** with Tempalte BART (Cui et al., 2021), a representative prompt-based learning method:

**Method Difference** Cui et al. (2021) follows an encoder-decoder framework. Given a text span, it decodes and *selects among all possible templates including none-entity templates*. Their candidates are templates based on a static set of entity types. In contrast, **BINDER** is to given an entity type *select among all possible text spans*. The candidates are a dynamic set of text spans, which are different for different input documents. The size of candidates is much larger – $\mathcal{O}(L^2)$, where $L$ is the max seq length, set to 256 in our experiments.

**Loss Difference** Cui et al. (2021) uses cross-entropy to *maximize the likelihood of gold entity type template against templates of other entity types*. In comparison, **BINDER** uses contrastive learning, given an entity type, to *maximize the similarity of gold text spans against other text spans*. Instead of comparing entity types, **BINDER** compares all possible text spans from an input document. It captures subtleties of each text span via contrastive learning. As shown in previous work, instance-based contrastive learning has better generalization performance than cross-entropy loss (Khosla et al., 2020), due to its robustness to noisy labels (Zhang & Sabuncu, 2018; Sukhbaatar et al., 2014) and the possibility of better margins (Elsayed et al., 2018). To our best knowledge, we are the first to demonstrate **BINDER** with contrastive loss (among text spans) significantly cross-entropy loss (among entity types/templates), in both supervised settings (see Table 1 and Table 2) and distantly supervised settings (see Table 3).

**Non-entity Handling Difference** Cui et al. (2021) labels all non-entity spans with non-entity templates. This can introduce false negatives when the training data is partially annotated (Das et al., 2022; Aly et al., 2021). Our formulation allows us to avoid using an explicit non-entity class, and instead to introduce a dynamic threshold based on the input document and the entity type, to distinguish entity spans from non-entity spans. Our experiments show a clear advantage of dynamic thresholding over explicit non-entity labeling scheme.

**Computational Efficiency** The encoder-decoder framework used by Cui et al. (2021) has much lower computational efficiency. First, it has double parameter size compared to **BINDER** (which is encoder-only). Second, both its training and inference time are significantly slower than **BINDER**. Because the decoding process relies on cross-attention from decoder to encoder, their method has to decode all possible templates for each input document. The number of templates $N_{\text{templates}} = N_{\text{text spans}} \times N_{\text{entity types}}$. And the overall the decoding operations are $\mathcal{O}(N_{\text{documents}} \times N_{\text{text spans}} \times N_{\text{entity types}})$. Furthermore, *the decoding process has to be done in an autoregressive manner*, which even reduces the time efficiency. In comparison, **BINDER** employs a bi-encoder framework. It separately encodes entity types and input documents. Encoding entity types does not rely on input documents. At test time, it only needs to encode entity types once, and then re-use them to for NER of different input documents. Encoding entity types is only $\mathcal{O}(N_{\text{entity types}})$, and can be done very efficiently in parallel on GPU.

**Performance on CoNLL03** To make a direct comparison with Cui et al. (2021), we train and evaluate **BINDER** on CoNLL03 (Tjong Kim Sang & De Meulder, 2003). Table 10 reports the test results. **BINDER** outperforms Template BART as well as other strong baselines.

Table 10: Test scores on CoNLL03.

|  | CoNLL03 | | |
| --- | --- | --- | --- |
|  | P | R | F1 |
| Yang et al. (2018) | - | - | 90.77 |
| Ma & Hovy (2016) | - | - | 91.21 |
| Gui et al. (2020) | - | - | 92.02 |
| Template BART (Cui et al., 2021) | 91.72 | 93.40 | 92.55 |
| **BINDER** | **93.08** | **93.57** | **93.33** |

## A.2 COMPARISON WITH SPAN-BASED NER

We highlight the difference between our method (**BINDER**) and span-based NER below:

**Formulation Difference** In span-based NER, it is to given a text span select among all entity types. The candidates are a static set of entity types, the size of which is usually small (e.g., `PER`, `ORG`, `LOC`, `MISC`). In contrast, **BINDER** formulates NER as given an entity type to *select among all possible text spans of an input document*. The candidates are a dynamic set of text spans, which are different for different input documents. The size of candidates is much larger $- \mathcal{O}(L^2)$, where $L$ is the max seq length, set to 256 in our experiments.

**Loss Difference**    Span-based NER uses cross-entropy to *maximize the likelihood of gold entity type template against templates of other entity types*. In comparison, **BINDER** uses contrastive learning, given an entity type, to *maximize the similarity of gold text spans against other text spans*. Instead of comparing entity types, **BINDER** compares all possible text spans from an input document. It captures subtleties of each text span via contrastive learning. As shown in previous work, instance-based contrastive learning has better generalization performance than cross-entropy loss (Khosla et al., 2020), due to its robustness to noisy labels (Zhang & Sabuncu, 2018; Sukhbaatar et al., 2014) and the possibility of better margins (Elsayed et al., 2018). To our best knowledge, we are the first to demonstrate **BINDER** with contrastive loss (among text spans) significantly cross-entropy loss (among entity types), in both supervised settings (see Table 1 and Table 2) and distantly supervised settings (see Table 3).

**Non-entity Handling Difference**    Span-based NER labels all non-entity tokens or spans with the same class `Outside` (`O`). This can introduce false negatives when the training data is partially annotated (Das et al., 2022; Aly et al., 2021). Our formulation allows us to avoid using an explicit non-entity class, and instead to introduce a dynamic threshold based on the input document and the entity type, to distinguish entity spans from non-entity spans. Our experiments show a clear advantage of dynamic thresholding over traditional explicit `O` labeling scheme.

### A.3    IMPACT OF MAXIMUM SEQUENCE LENGTH

In our experiments, we set the maximum sequence length $L$ to 256, meaning that the number of candidate spans is $\mathcal{O}(L^2)$. Speed comparison in Table 7 is made based on $L = 256$. We also experimented with smaller numbers of $L$ (e.g., 64, 128), which resulted in better speed but a slight performance degradation. Increasing $L$ did not bring significant gain but increased memory consumption. Therefore, in the experiments, we set $L$ to 256.

### A.4    IMPLEMENTATION DETAILS

We implement our models based on the HuggingFace Transformers library (Wolf et al., 2020). The base encoders are initialized using PubMedBERT-base-uncased (Gu et al., 2021) or BioBERT (Lee et al., 2019) for biomedical NER datasets, and BERT-base-uncased or BERT-large-uncased (Devlin et al., 2019) for NER datasets in the general domains. The linear layer output size is 128; the width embedding size is 128; the initial temperatures are 0.07. We train our models with the AdamW optimizer (Loshchilov & Hutter, 2017) of a linear scheduler and dropout of 0.1. The entity start/end/span contrastive loss weights are set to $\alpha = 0.2, \gamma = 0.2, \lambda = 0.6$, and the same loss weights are chosen for thresholding contrastive learning. The contrastive losses for thresholding and entity are weighted equally in the final loss. For all experiments, we ignore sentence boundaries, and tokenize and split text into sequences with a stride of 16. For base encoders, we train our models for 20 epochs with a learning rate of 3e-5 and a batch size of 8 sequences with the maximum token length of $N = 128$. For large encoders, we train our models for 40 epochs with a learning rate of 3e-5 and a batch size of 16 sequences with the maximum token length of $N = 256$. The maximum token length for entity spans is set to 30. We use early stop with a patience of 10 in the distantly supervised setting. Validation is done at every 50 steps of training, and we adopt the models that have the best performance on the development set. We report the median score of multiple runs.

### A.5    INFERENCE FOR **BINDER**

As we can see in Algorithm 1, the difference between joint position-span and span-only strategies is whether line 9 is used. Also, for flat NER datasets, we further carry out a post-processing step to remove overlapping predictions (line 19 in Algorithm 1). Here, the post-processing (`removeOverlap`) is carried out in a greedy fashion where higher scored span predictions with earlier start and end positions are preferred.

---

**Algorithm 1:** Inference for **BINDER**.

---

**Input:** $\mathcal{S} = \{(i,j)|i,j = 1,\ldots,N, 0 \le j - i \le L\}$ the set of spans , $\mathcal{E} = \{E_1,\ldots,E_K\}$ the set of entity types, `joint` for whether using joint position-span inference, and `flat` for whether the inference is for flat NER.

$M = \{\}$;
**for** $E_k \in \mathcal{E}$ **do**

4      compute start/end/span threshold scores $b_{null}, e_{null}, s_{null}$.

6      **for** $(i,j) \in \mathcal{S}$ **do**

         compute start/end/span similarity scores $b, e, s$.

9          **if** `joint` *is true and* $b < b_{null}$ *or* $e < e_{null}$ **then**

            Continue;

         **end**

13          **if** $s > s_{null}$ **then**

            $M = M \bigcup \{(i,j,E_k)\}$;

         **end**

     **end**

**end**

19 **if** *flat is true* **then**

     **return** `removeOverlap(M)`;

**end**

**return** $M$;

**Function** `removeOverlap(`$\hat{D}$`)`:

     $\hat{M} = \{\}$;

     sort $\hat{D}$ by the similarity score in descending order and break the tie by ascending in start and end positions;

     **for** $(i,j,E_k)$ *in* $\hat{D}$ **do**

         **if** *span* $(i,j)$ *has no overlap in* $\hat{M}$ **then**

            $\hat{M} = \hat{M} \cup \{(i,j,E_k)\}$;

         **end**

     **end**

     **return** $\hat{M}$;

---

## A.6   STATISTICS OF DATASETS

Table 11 reports the statistics of supervised NER datasets.

Table 11: The statistics of supervised NER datasets.

| Dataset | $|\mathcal{E}|$ | # Annotations | | |
|---|---|---|---|---|
| | | Train | Dev | Test |
| ACE2004 | 7 | 22,735 (5-fold) | | |
| ACE2005 | 7 | 26,473 | 6,338 | 5,476 |
| GENIA | 5 | 46,142 | 4,367 | 5,506 |
| BC5-chem | 1 | 5,203 | 5,347 | 5,385 |
| BC5-disease | 1 | 4,182 | 4,244 | 4,424 |
| NCBI | 1 | 5,134 | 787 | 960 |
| BC2GM | 1 | 15,197 | 3,061 | 6,325 |
| JNLPBA | 1 | 46,750 | 4,551 | 8,662 |

## A.7   DISTRIBUTIONS OF SIMILARITY SCORES

Figure 2 visualizes the distributions of similarity scores between different text spans and entity types based on different thresholding strategies. Consistent with the scores in Table 6, the majority of entity spans and non-entity spans are separable regardless of the thresholding strategy. This is true even when no thresholds are used during training and instead we tune global thresholds on dev.

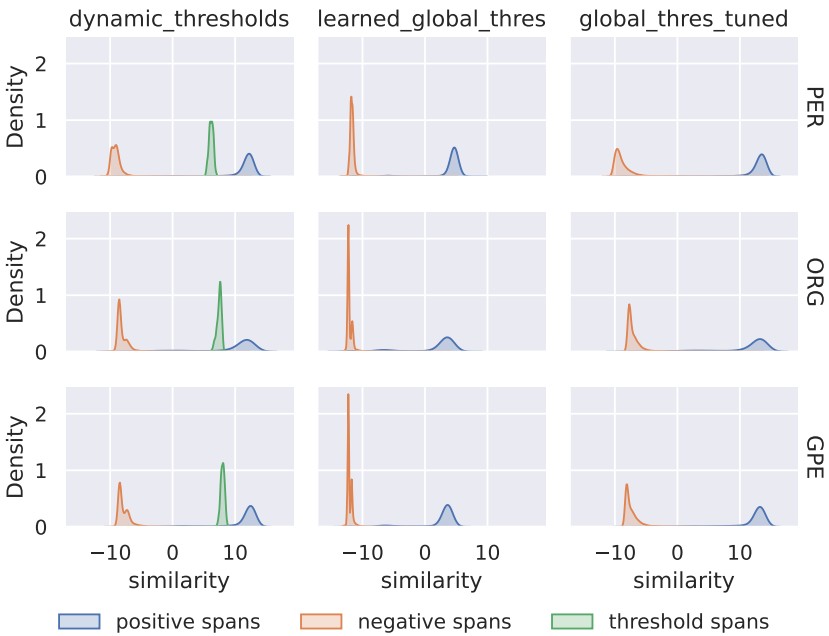

Figure 2: The kernel density estimation of similarity scores between different text spans (entity, non-entity, and threshold spans) and entity types (`PER`, `ORG`, `GPE`) on ACE2005 based on different thresholding strategies.

We zoom in the distributions in Figure 3 and observe that the learned global thresholds tend to make the similarities less separable between entity spans and non-entity spans.

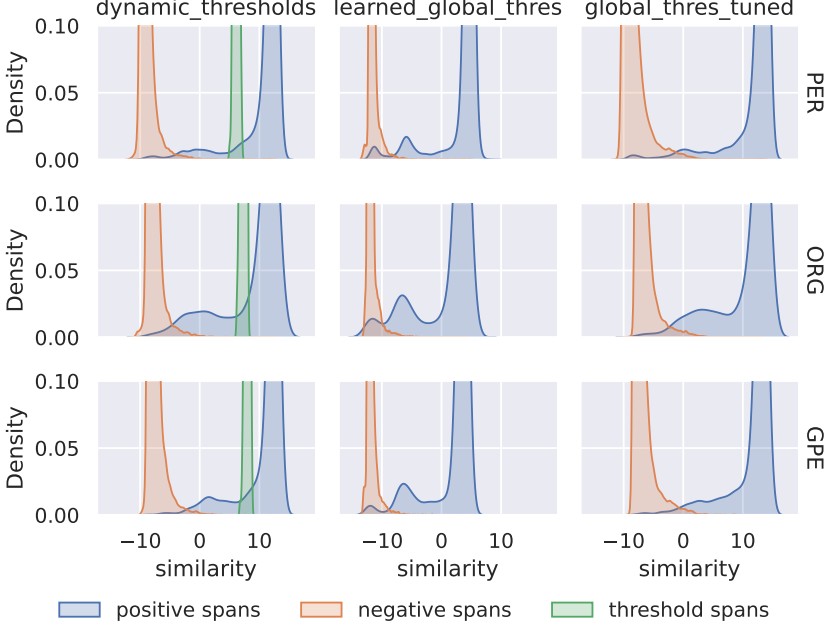

Figure 3: Zoom-in of the kernel density estimation of similarity scores in Figure 2, with the $y$-axis density limited to $(0, 0.1)$.

