# OpenReview forum: "Optimizing Bi-Encoder for Named Entity Recognition via Contrastive Learning"
_ICLR.cc/2023/Conference — ICLR 2023 poster_

### Official Review · Reviewer_QTaJ · 2022-10-25

**Confidence:** 4
**Correctness:** 3
**Technical Novelty And Significance:** 3
**Empirical Novelty And Significance:** 3
**Recommendation:** 5

**Clarity, Quality, Novelty And Reproducibility:**

Some important details are missed or not cleared. For example, how to set the maximum length L. What is the relation between L and the model performance (accuracy and speed).  I didn't see the open source statement in this paper, as the experiments require many details, without the source code, it is hard to reproduce the results.

**Strength And Weaknesses:**

Some important details are missed or not cleared. For example, how to set the maximum length L. What is the relation between L and the model performance (accuracy and speed). Experiments didn't include the most common dataset CoNLL 03. This framework setting is similar to the template-based NER (using prompt,"Template-based named entity recognition using BART") but this work didn't compare with this work.

**Summary Of The Paper:**

This paper proposed a contrastive learning framework for NER tasks. The proposed model is flexible in predicting nested entities.  The idea is novel and experiments demonstrate the effectiveness of the proposed model.

**Summary Of The Review:**

Novel and strong results, but the details are not complete and the experiments/comparison are not complete.

---

> ### Author Response · Authors · 2022-11-06
> **Response to Reviewer QTaJ**
>
> - Thanks for your review! We want to highlight the difference between our method (Binder) and "Template-based named entity recognition using BART", Cui et al. (2021) below (also see `Appendix A.1` in our revision):
>     - **Formulation Difference**: Cui et al. (2021) follows an encoder-decoder framework. Given a text span, it decodes and *selects among all possible templates including none-entity templates*. Their candidates are templates based on a static set of entity types. In contrast, our method (Binder) is to given an entity type *select among all possible text spans*. The candidates are a dynamic set of text spans, which are different for different input documents. The size of candidates is much larger -- $\mathcal{O}(L^2)$, where $L$ is the max seq length, set to 256 in our experiments.
>     - **Loss Difference**: Cui et al. (2021) uses cross-entropy to *maximize the likelihood of gold entity type template against templates of other entity types*. In comparison, Binder uses contrastive learning, given an entity type, to *maximize the similarity of gold text spans against other text spans*. Instead of comparing entity types, Binder compares all possible text spans from an input document. It captures subtleties of each text span via contrastive learning. As shown in previous work, instance-based contrastive learning has better generalization performance than cross-entropy loss ([Khosla et al., 2021](https://arxiv.org/abs/2004.11362)), due to its robustness to noisy labels ([Sukhbaatar et al., 2015](https://arxiv.org/abs/1406.2080)) and the possibility of better margins ([Elsayed et al., 2018](https://arxiv.org/abs/1803.05598))). To our best knowledge, we are the first to demonstrate Binder with contrastive loss (among text spans) significantly cross-entropy loss (among entity types/templates), in both supervised settings and distantly supervised settings.
>     - **Non-entity Handling Difference**: Cui et al. (2021) labels all non-entity spans with non-entity templates. This can introduce false negatives when the training data is partially annotated ([Das et al., 2022](https://aclanthology.org/2022.acl-long.439/); [Aly et al., 2021](https://aclanthology.org/2021.acl-long.120/)). Our formulation allows us to avoid using an explicit non-entity class, and instead to introduce a dynamic threshold based on the input document and the entity type, to distinguish entity spans from non-entity spans. Our experiments show a clear advantage of dynamic thresholding over explicit non-entity labeling scheme.
>     - **Computational Efficiency**: The encoder-decoder framework used by Cui et al. (2021) has much lower computational efficiency. First, it has doubled parameter size compared to Binder (which is encoder-only). Second, both its training and inference time are significantly slower than Binder. Because the decoding process relies on cross-attention from decoder to encoder, their method has to decode all possible templates for each input document. The number of templates $N_\textrm{templates}=N_\textrm{text spans} \times N_\textrm{entity types}$. And the overall the decoding operations are $\mathcal{O}(N_\textrm{documents} \times N_\textrm{text spans} \times N_\textrm{entity types})$. Furthermore, **the decoding process has to be done in an autoregressive manner**, which even reduces the time efficiency. In comparison, Binder employs a bi-encoder framework. It separately encodes entity types and input documents. Encoding entity types does not rely on input documents. At test time, it only needs to encode entity types once, and then re-use them to for NER of different input documents. Encoding entity types is only $\mathcal{O}(N_\textrm{entity types})$, and can be done very efficiently in parallel on GPU.
>     - **Performance on CoNLL03**: We added a direct comparison w. Cui et al. (2021) on CoNLL03 in `Appendix Tbl 10` and the able below - we achieve 93.33 F1, outperforming their best reported number 92.55 F1.
> |  | P | R | F1 |
> |---|:---:|:---:|:---:|
> | Yang et al. (2018) | - | - | 90.77 |
> | Ma & Hovy (2016) | - | - | 91.21 |
> | Gui et al. (2020) | - | - | 92.02 |
> | Cui et al. (2021) | 91.72 | 93.40 | 92.55 |
> | **Binder (Ours)** | **93.08** | **93.57** | **93.33** |
> - Re **max seq length** $L$, we stated in `Appendix A.4` that we set $L$ to 256, meaning the number of candidate spans is $\mathcal{O}(L^2)$. Speed comparison in Tbl 7 is based on $L=256$. We also experimented with smaller numbers of $L$ (e.g., 64, 128), which resulted in better speed but a slight performance degradation. Increasing $L$ didn't bring significant gain but increased memory consumption. Therefore, in the experiments, we set $L$ to 256. We added `Appendix A.3` to describe the impact of max seq length.
> - Re **reproducibility**, please check out the updated supplemental material. We added an anonymous version of our codebase, including README to run the code. We will release the codebase to the public as well.
>
> Thanks again for your review!

---

### Official Review · Reviewer_FXLT · 2022-10-25

**Confidence:** 4
**Correctness:** 3
**Technical Novelty And Significance:** 3
**Empirical Novelty And Significance:** 3
**Recommendation:** 6

**Clarity, Quality, Novelty And Reproducibility:**


- The overall writing is clear.
- The quality meets the standard of ICLR.
- The novelty is relatively limited.
- As the paper has some detailed designs on special losses, the code is suggested to be public to reproduce empirical reesults.

**Strength And Weaknesses:**


Strength:
- Utilizing Bi-encoder has never been considered, while this idea is very similar to the span-based ner. (see Weaknesses)
- The empirical results are extensive.

Weaknesses:
- The proposed approach is very similar to span-based NER. In span-based NER, the label type embeddings are randomly initialized (the final layer before softmax). However, the proposed approach utilizes a neural network to obtain that matrix. In the end, the novelty seems limited.
- Besides nested NER, I would also prefer adding some flat NER results (CoNLL, WNUT, and so on).

**Summary Of The Paper:**

This paper proposes a new framework that performs the Named Entity Recognition task from the semantic matching perspective (bi-encoder here). The idea is to map candidate text spans and entity types into the same vector representation space and perform the NER task using distance metric accordingly. One important issue is separating non-entity spans from desired entity mentions. To alleviate these problems, the paper proposes a position-based objective and a dynamic thresholding approach. Experiments on both supervised and distantly supervised NER tasks demonstrate the effectiveness of the proposed approaches.

**Summary Of The Review:**

This paper proposes a new framework that performs the Named Entity Recognition task with the bi-encoder model. The paper shows strong results on nested NER and distant supervised NER tasks. While the results are promising, the proposed approach is similar to the span-based NER approach, which might have limited techniqual novelty.

---

> ### Author Response · Authors · 2022-11-06
> **Response to Reviewer FXLT**
>
> - Thanks for your review! We want to highlight the difference between our method (Binder) and span-based NER below (also see `Appendix A.2` in our revision):
>     - **Formulation Difference**: In span-based NER, it is to given a text span select among all entity types. The candidates are a static set of entity types, the size of which is usually small (e.g., PER, ORG, LOC, MISC). In contrast, Binder formulates NER as given an entity type to *select among all possible text spans of an input document*. The candidates are a dynamic set of text spans, which are different for different input documents. The size of candidates is much larger -- $\mathcal{O}(L^2)$, where $L$ is the max seq length, set to 256 in our experiments.
>     - **Loss Difference**: Span-based NER uses cross-entropy to maximize the likelihood of gold entity type against other entity types. In comparison, Binder uses contrastive learning, given an entity type, to *maximize the similarity of gold text spans against other text spans*. Instead of comparing a small set of entity types, Binder compares all possible text spans from an input document. It captures subtleties of each text span via contrastive learning. As shown in previous work, instance-based contrastive learning has better generalization performance than cross-entropy loss ([Khosla et al., 2021](https://arxiv.org/abs/2004.11362)), due to its robustness to noisy labels ([Sukhbaatar et al., 2015](https://arxiv.org/abs/1406.2080)) and the possibility of better margins ([Elsayed et al., 2018](https://arxiv.org/abs/1803.05598))). To our best knowledge, we are the first to demonstrate Binder with contrastive loss (among text spans) significantly cross-entropy loss (among entity types/templates), in both supervised settings and distantly supervised settings.
>     - **Non-entity Handling Difference**: Span-based NER (or generally classification-based NER) labels all non-entity tokens or spans as the same class $\texttt{Outside}$ ($\texttt{O}$). This can introduce false negatives when the training data is partially annotated ([Das et al., 2022](https://aclanthology.org/2022.acl-long.439/); [Aly et al., 2021](https://aclanthology.org/2021.acl-long.120/)). Our formulation allows us to avoid using an explicit non-entity class, and instead to introduce a dynamic threshold based on the input document and the entity type to distinguish entity spans from non-entity spans. Our experiments show a clear advantage of dynamic thresholding over traditional explicit $\texttt{O}$-labeling scheme.
> - Re flat NER, we added `Appendix Tbl 10` to compare Binder with the recent prompt-based NER as well as several strong baselines on CoNLL03. We also pasted the table below:
> |                                  |     P     |     R     |     F1    |
> |----------------------------------|:---------:|:---------:|:---------:|
> | [Yang et al. (2018)](https://aclanthology.org/C18-1327/)               |     -     |     -     |   90.77   |
> | [Ma & Hovy (2016)](https://aclanthology.org/P16-1101/)                 |     -     |     -     |   91.21   |
> | [Gui et al. (2020)](https://aclanthology.org/2020.emnlp-main.181/)                |     -     |     -     |   92.02   |
> | Template BART ([Cui et al., 2021](https://aclanthology.org/2021.findings-acl.161/)) |   91.72   |   93.40   |   92.55   |
> | **Binder** (Ours)                    | **93.08** | **93.57** | **93.33** |
> - Re reproducibility, **we will release the fully reproducible codebase. We added an anonymous version of our codebase in the supplemental material, where we include README for reviewers to run the code.**
>
> Thanks again for your review!

---

### Official Review · Reviewer_wC16 · 2022-10-28

**Confidence:** 4
**Correctness:** 4
**Technical Novelty And Significance:** 3
**Empirical Novelty And Significance:** 3
**Recommendation:** 8

**Clarity, Quality, Novelty And Reproducibility:**

Clarity: Good as the paper is easy to follow. The paper still can be further polished with proofreading.
Quality: Good. The technique part is sound and solid. Extensive experiments and ablations are diverse and insightful.
Novelty: Good. The authors formulate the problem under contrastive learning with new methodology, though these exist some works are relevant to this work closely.
Reproducibility: No data point as the code is not shared.

**Strength And Weaknesses:**

Strength
- This work is well-motivated. The idea of framing NER as representation learning is interesting. Although there are some works combine contrastive learning with NER, this paper proposed novel strategies to handle challenges specifically.
- The introduced position-based objective to treat entity spans differently. Specifically, they penalize more entity span that have completely no overlap with the gold span while penalize less partially correct span. Their experiment results (as shown in Table 8) implies this objective contribute performance gains consistently and substantially.
- The authors conducted extensive experiment in various settings and benchmarks. Their methods beats SOTA baselines almost across the board.
- Various ablation studies have been conducted to share insightful findings.

Major Concerns
- There are serval other works [1-3] which are quite relevant/similar with this paper as they solve NER under contrastive learning framework as well. I would appreciate the authors if they can compare their method with these works from methodology perspective.
- I am wondering how would the model handle entity name that could be a valid value in different entity types. For instance, Willson could be a person name, while it also could be a ORG(company) name. Could you share some thoughts on this case?
- Currently the method enumerates all candidate entity spans as negative samples for calculating contrastive losses. In the paper, the maximum token length is 256. However if we have even larger sequence length and/or large $L$, do we expect the computational cost would increase significantly? If so, is there any alternative and more efficient way to create negative samples?
- For span-based objective, what is the dimension of $h_i^T$? I guess it should be $\mathbb{R}^m$? Could you elaborate more about $D$? Is it learnable? and how to learn it? Is $D(j-i)$ refers to the token representations between $h_i^T$ and $h_j^T$? It seems the answer is not. So why do you using the token embeddings of $h_i^T$ and $h_j^T$ only instead of taking token embeddings between them into consideration?
- In eq(8), is $u_{[CLS]}=Linear_B^T(h_{[CLS]}^T)$? Do you think $u_{[CLS]}$ would collapse to $e_k^B$? why or why not?
- In eq(10), what is $\ell_{\text{joint}}^+$? Are you referring to $\ell_{span}^+$?

[1]. Fu, Yingwen, Nankai Lin, Ziyu Yang and Shengyi Jiang. “A Dual-Contrastive Framework for Low-Resource Cross-Lingual Named Entity Recognition.” ArXiv abs/2204.00796 (2022): n. pag.
[2]. Ye, Hongbin, Ningyu Zhang, Shumin Deng, Mosha Chen, Chuanqi Tan, Fei Huang and Huajun Chen. “Contrastive Triple Extraction with Generative Transformer.” AAAI (2021).
[3]. Lin, Bill Yuchen, Dong-Ho Lee, Minghan Shen, Ryan Rene Moreno, Xiao Huang, Prashant Shiralkar and Xiang Ren. “TriggerNER: Learning with Entity Triggers as Explanations for Named Entity Recognition.” ACL (2020).

**Summary Of The Paper:**

Most of existing works approach NER as sequence labeling, span classification, or sequence-to-sequence problems. These conventional methods rely on CRF to extract entity values which performs subpar on nested NER tasks. To address these limitations, this work formulate NER as a representation learning problem. Intuitively, their method associate entity type embeddings with entity name embeddings in a common subspace. To achieve the goal in a principle manner, they cast the problem into contrastive learning framework. Moreover, they proposed two contrastive objectives for model training and dynamic thresholding for adaptive model inference. Extensive experiment demonstrated the effectiveness of their approach and ablations conveyed insightful findings.

**Summary Of The Review:**

Overall the proposed method for NER is moderately novel. It can benefit the community in this filed. The experiments and ablations demonstrated the effectiveness of their method.

---

> ### Author Response · Authors · 2022-11-07
> **Response to Reviewer wC16**
>
> - Thank you for your thorough understanding of our paper and thoughtful review! We compare with those related papers below:
>     - "TriggerNER: Learning with Entity Triggers as Explanations for Named Entity Recognition" assumes the availability of trigger annotations. They apply not only contrastive learning between triggers and sentences, but also cross-entropy loss to trigger type classification. Their model needs a two-stage training: they first jointly train the TrigEncoder and TrigMatcher, and then re-use the first stage output to train a sequence tagger. In comparison, our method (Binder) only requires a single-stage training. Binder does not assume extra annotations. The learning process of Binder is only based on contrastive loss.
>     - "A Dual-Contrastive Framework for Low-Resource Cross-Lingual Named Entity Recognition" also applies contrastive loss and cross-entropy loss at the same time. Ablation study in Tbl 4 of their paper shows that cross-entropy loss plays the major role: Without contrastive loss, their model only has a minor performance decrease. In contrast, contrastive loss is the only loss used in Binder. It plays a vital role. Another difference is that their contrastive loss compares tokens/sentences from different examples, whereas Binder uses contrastive learning to capture the subtleties of text spans from the same examples.
>     - "Contrastive Triple Extraction with Generative Transformer" follows an encoder-decoder framework. But it is designed for end-to-end relation extraction, not NER. We compare with a similar paper "Template-based Named Entity Recognition using BART" Cui et al. (2021), which applies an encoder-decoder framework for NER. Please see `Appendix A.1` in our revision for the comparison.
> - Entity name that could be a valid value in different entity types is a typical case that requires disambiguation. We resolve these ambiguous entity names by reading their context. Specifically, Binder generates contextual encodings via a pretrained language model and learn the subtleties of these similar text spans via contrastive loss. Tbl 1 and Tbl 2 in our experiments show that Binder is indeed better than existing methods at disambiguation.
> - Re max seq length $L$, we set it to 256, meaning the number of candidate spans is $\mathcal{O}(L^2)$. We also experimented with smaller numbers of  $L$ (e.g., 64, 128), which resulted in better speed but a slight performance degradation. Increasing $L$ didn't bring significant gain but did increase memory consumption. Therefore, we set it to 256. We added `Appendix A.3` to describe the impact of max seq length. An alternative to increase efficiency of negative sampling would be to introduce latent modeling of triggers as the TriggerNER paper. We would like to explore it as a future direction.
> - $h_i^T \in \mathbb{R}^d$. $D\in\mathbb{R}^{L\times m}$ is a learnable embedding matrix. $D(j-i)\in\mathbb{R}^m$ is the ($j −i$)-th row of $D$. $D(j-i)$ refers to the embedding for spans whose width is $j-i$. $D$ is learned end-to-end as the other parameters. We only use $h_i^T$ and $h_j^T$ to represent the span for computational efficiency. Please check out line 257-264 in `src/model.py` in the updated supplemental material. Essentially, it is `torch.cat` of sequence encodings from different views. This is done efficiently in parallel on GPU.
> - Yes, $\mathbf{u}_\texttt{[CLS]}=\texttt{Linear}^T_B(\mathbf{h}^T_\texttt{[CLS]})$, $\mathbf{e}^B_k=\texttt{Linear}^E_B(\mathbf{h}^{E_k}_\texttt{[CLS]})$. Note that $\mathbf{h}^{E_k}_\texttt{[CLS]}$ and $\mathbf{h}^T_\texttt{[CLS]}$ are from two different encoders, which do not share parameters. Therefore, $\mathbf{u}_\texttt{[CLS]}$ do not collapse to $\mathbf{e}^B_k$ at the beginning of training. During training, eq(6) pulles ($\mathbf{u}_\texttt{[CLS]}$, $\mathbf{e}^B_k$) apart, because they are in the denominator. On the other hand, Eq(8) pushes ($\mathbf{u}_\texttt{[CLS]}$, $\mathbf{e}^B_k$) closer. Eventually $\mathbf{u}_\texttt{[CLS]}$ is placed at the boundary to tell entity spans from non-entity spans, i.e., dynamic thresholding.
> - Yes, $\ell^+_\textrm{joint}$ refers to $\ell^+_\textrm{span}$. We've fixed this typo in the revision.
> - Re **reproducibility**, please check out the updated supplemental material, where we added an anonymous codebase, including README for reviewers to run the code. We will release the fully reproducible codebase to the public.
>
> Thanks again for your review!

---

> > ### Comment · Reviewer_wC16 · 2022-11-23
> > **Concerns addressed**
> >
> > Thanks for the clarification and the codebase for reproducibility. My concerns have been well addressed. I am looking forward to read the final version.

---

### Official Review · Reviewer_mFSQ · 2022-11-02

**Confidence:** 4
**Correctness:** 2
**Technical Novelty And Significance:** 2
**Empirical Novelty And Significance:** 2
**Recommendation:** 6

**Clarity, Quality, Novelty And Reproducibility:**

Writing is unclear.
- What is D(j-I). Span width embedding matrix?
- What is l_joint?

Not clear about the reproducibility.


**Strength And Weaknesses:**

Paper presents bi-encoder style NER framework with novel objectives suitable to NER and shows its effectiveness in nested NER problems. However, I have following major concerns:

(1) Lack of baseline. Span-based NER, which predicts over all the candidate spans in the sentence, has been widely explored in prompt-based learning (Ding et al., 2021, Cui et al., 2021, Ma et al., 2022… etc.) and . It should have baselines from other research lines for making this paper be more convincing.

(2) No details about candidate generation. Based on window-size, I think the number of span candidates will vary a lot. This ambiguity makes Table 7 be not convincing since the inference speed might be affected by the number of span candidates.

Prompt-Learning for Fine-grained Entity Typing,, Ding et al., 2021
Template-based Named Entity Recognition using BART, Cui et al., 2021
Template-free Prompt Tuning for Few-shot NER., Ma et al., 2022


**Summary Of The Paper:**

Paper presents a bi-encoder framework for NER, which applies contrastive learning to map candidate text spans and entity types into the same vector representation space and make the representation of entity mentions be similar with corresponding entity type. It proposes span-based objectives that compare span and entity type, and position-based objectives that compare start and end tokens separately with entity type. It shows its effectiveness in nested NER with fast inference speed.

**Summary Of The Review:**

Paper presents bi-encoder style NER framework with novel objectives suitable to NER and shows its effectiveness in nested NER problems. However, I think paper needs to be improved a lot in terms of writing and experiments.

---

> ### Author Response · Authors · 2022-11-06
> **Response to Reviewer mFSQ**
>
> - Thanks for pointing out the related work on prompt-based learning (Ding et al., 2021; Cui et al., 2021; Ma et al., 2022; inter alia). In the revision, we have made a direct comparison w. this line of work in `Appendix A.1`. Using Cui et al. (2021) as a representative method, we would like to emphasize the difference below:
>     -  **Formulation Difference**: Cui et al. (2021) follows an encoder-decoder framework. Given a text span, it decodes and *selects among all possible templates including none-entity templates*. Their candidates are templates based on a static set of entity types. In contrast, our method (Binder) is to given an entity type *select among all possible text spans*. The candidates are a dynamic set of text spans, which are different for different input documents. The size of candidates is much larger -- $\mathcal{O}(L^2)$, where $L$ is the max seq length, set to 256 in our experiments.
>     - **Loss Difference**: Cui et al. (2021) uses cross-entropy to *maximize the likelihood of gold entity type template against templates of other entity types*. In comparison, Binder uses contrastive learning, given an entity type, to *maximize the similarity of gold text spans against other text spans*. Instead of comparing entity types, Binder compares all possible text spans from an input document. It captures subtleties of each text span via contrastive learning. As shown in previous work, instance-based contrastive learning has better generalization performance than cross-entropy loss ([Khosla et al., 2021](https://arxiv.org/abs/2004.11362)), due to its robustness to noisy labels ([Sukhbaatar et al., 2015](https://arxiv.org/abs/1406.2080)) and the possibility of better margins ([Elsayed et al., 2018](https://arxiv.org/abs/1803.05598))). To our best knowledge, we are the first to demonstrate Binder with contrastive loss (among text spans) significantly cross-entropy loss (among entity types/templates), in both supervised settings and distantly supervised settings.
>     - **Non-entity Handling Difference**: Cui et al. (2021) labels all non-entity spans with non-entity templates. This can introduce false negatives when the training data is partially annotated ([Das et al., 2022](https://aclanthology.org/2022.acl-long.439/); [Aly et al., 2021](https://aclanthology.org/2021.acl-long.120/)). Our formulation allows us to avoid using an explicit non-entity class, and instead to introduce a dynamic threshold based on the input document and the entity type, to distinguish entity spans from non-entity spans. Our experiments show a clear advantage of dynamic thresholding over explicit non-entity labeling scheme.
>     - **Computational Efficiency**: The encoder-decoder framework used by Cui et al. (2021) has much lower computational efficiency. First, it has doubled parameter size compared to Binder (which is encoder-only). Second, both its training and inference time are significantly slower than Binder. Because the decoding process relies on cross-attention from decoder to encoder, their method has to decode all possible templates for each input document. The number of templates $N_\textrm{templates}=N_\textrm{text spans} \times N_\textrm{entity types}$. And the overall the decoding operations are $\mathcal{O}(N_\textrm{documents} \times N_\textrm{text spans} \times N_\textrm{entity types})$. Furthermore, **the decoding process has to be done in an autoregressive manner**, which even reduces the time efficiency. In comparison, Binder employs a bi-encoder framework. It separately encodes entity types and input documents. Encoding entity types does not rely on input documents. At test time, it only needs to encode entity types once, and then re-use them to for NER of different input documents. Encoding entity types is only $\mathcal{O}(N_\textrm{entity types})$, and can be done very efficiently in parallel on GPU.
>     - **Performance on CoNLL03**: To make a direct comparison w. Cui et al. (2021), we added our result on CoNLL03 - we achieve **93.33** F1, outperforming their best reported number 92.55 F1. Please check out the full table in the next response.
> - Re **candidate generation**, please check out line 207-213 in `src/model.py` in the updated supplemental material. Essentially, it is `torch.cat` of sequence encodings from different views. This is done efficiently in parallel on GPU. The number of candidate spans only depends on the max seq length $L$ of text encoder. We stated in `Appendix A.4` $L=256$, meaning the number of candidates is $\mathcal{O}(L^2)$. We added section in `Appendix A.3` to describe the impact of $L$.
> - $D(j-i)$ is the learnable embedding for spans with width of $j-i$.
> - $\ell_\textrm{joint}$ is a typo. It refers to $\ell_\textrm{span}$. We've fixed it in the revision.
> - Re reproducibility, **We have added an anonymous version of codebase in the supplemental material, including README to run the code.** We will release the codebase to the public as well.

---

> > ### Author Response · Authors · 2022-11-15
> > **Comparison w. Cui et al. (2021) on CoNLL03**
> >
> > The table below reports the test results on CoNLL03. Binder outperforms Template BART ([Cui et al., 2021](https://aclanthology.org/2021.findings-acl.161/)) as well as other strong baselines:
> > |                                  |     P     |     R     |     F1    |
> > |----------------------------------|:---------:|:---------:|:---------:|
> > | [Yang et al. (2018)](https://aclanthology.org/C18-1327/)               |     -     |     -     |   90.77   |
> > | [Ma & Hovy (2016)](https://aclanthology.org/P16-1101/)                 |     -     |     -     |   91.21   |
> > | [Gui et al. (2020)](https://aclanthology.org/2020.emnlp-main.181/)                |     -     |     -     |   92.02   |
> > | Template BART ([Cui et al., 2021](https://aclanthology.org/2021.findings-acl.161/)) |   91.72   |   93.40   |   92.55   |
> > | **Binder** (Ours)                    | **93.08** | **93.57** | **93.33** |

---

> > > ### Comment · Reviewer_mFSQ · 2022-11-21
> > > **RE: Response**
> > >
> > > Thank you for reflecting my review and for further experimentation.
> > > I just raised my score to 6. Most of writing concerns are solved and I could understand your approach more.
> > >
> > > I have few more comments.
> > >
> > > 1. I recommend testing bidirectional encoder-only models (e.g. BERT, RoBERTa) for span-based NER rather than encoder-decoder model structures (e.g. BART) as this is a fairer comparison.
> > >
> > > 2. I want to see the effect of data set size. While contrast learning seems to be effective, there are minor concerns that it is only effective on large training data settings.
> > >
> > > Thanks again for your thorough experiments.

---

### Decision · Program_Chairs · 2023-01-20

**Decision:**

Accept: poster

**Justification For Why Not Higher Score:**

While the work is solid, I think the topic is a little narrow – it is on named entity recognition only. Therefore, perhaps it is worthwhile to make it a poster presentation. This way, those who are particularly interested in the topic may be able to have closer interactions with the authors during the presentation.

**Justification For Why Not Lower Score:**

This submission presents compelling findings that are worth sharing. I don't see a reason to reject it.

**Metareview: Summary, Strengths And Weaknesses:**

This paper proposes a novel approach to the problem of named entity recognition by casting it as a representation learning task. Through contrastive learning, the model learns to distinguish spans that are valid entities from those that do not form entities.

The paper is making an exciting contribution to the field of named entity recognition, which may change many researchers' views of such a task. Backed by solid empirical results on both flat entity and nested entity recognition tasks, the work presents robust research findings worth sharing with the community.

Several reviewers requested additional experiments and raised questions on further details (such as the impact of L). The authors provided thorough information in the rebuttals.

**Note From Pc:**

if the above contains the word "oral" or "spotlight" please see: "oral" presentation means -> notable-top-5% and "spotlight" means -> notable-top-25%. As stated in our emails, we are disassociating presentation type from AC recommendations